# Intention Prediction of a Hypersonic Glide Vehicle Using a Satellite Constellation Based on Deep Learning

**Yu Cheng, Cheng Wei \*, Yongshang Wei, Bindi You and Yang Zhao**

School of Aeronautics, Harbin Institute of Technology, Harbin 150006, China
\* Correspondence: weicheng@hit.edu.cn; Tel.: +86-186-4656-2866

**Abstract:** Tracking of hypersonic glide vehicles (HGVs) by a constellation tracking and observation system is an important part of the space-based early warning system. The uncertainty in the maneuver intentions of HGVs has a non-negligible impact on the tracking and observation process. The cooperative scheduling of multiple satellites in an environment of uncertainty in the maneuver intentions of HGVs is the main problem researched in this paper. For this problem, a satellite constellation tracking decision method that considers the HGVs' maneuver intentions is proposed. This method is based on building an HGV maneuver intention model, developing a maneuver intention recognition and prediction algorithm, and designing a sensor-switching strategy to improve the local consensus-based bundle algorithm (LCBBA). Firstly, a recognizable maneuver intention model that can describe the maneuver types and directions of the HGVs in both the longitudinal and lateral directions was designed. Secondly, a maneuver intention recognition and prediction algorithm based on parallel, stacked long short-term memory neural networks (PSLSTM) was developed to obtain maneuver directions of the HGV. On the basis of that, a satellite constellation tracking decision method (referred to as SS-LCBBA in the following) considering the HGVs' maneuver intentions was designed. Finally, the maneuver intention prediction capability of the PSLSTM network and two currently popular network structures: the multilayer LSTM (M-LSTM) and the dual-channel and bidirectional neural network (DCBNN) were tested for comparison. The simulation results show that the PSLSTM can recognize and predict the maneuver directions of HGVs with high accuracy. In the simulation of a satellite constellation tracking HGVs, the SS-LCBBA improved the cumulative tracking score compared to the LCBBA, the blackboard algorithm (BM), and the variable-center contract network algorithm (ICNP). Thus, it is concluded that SS-LCBBA has better adaptability to environments with uncertain intentions in solving multi-satellite collaborative scheduling problems.

**Keywords:** satellite constellation tracking system; hypersonic glide vehicles; maneuver intention prediction; sensor-switching strategy; parallel stacking neural network

**MSC:** 90C39





## 1. Introduction

### 1.1. Satellite Constellation Tracking System

Due to the capacity to continuously track and observe moving targets in space across the entire airspace, satellite constellation tracking systems have gained considerable interest in recent years [1]. The U.S. Space Development Agency's (SDA) draft solicitation mentioned a global missile tracking space sensor satellite constellation of 28 satellites. The satellite constellation tracking system is composed of a group of low-orbiting satellites carrying infrared tracking sensors. Each satellite tracks moving targets in space via a narrow-view infrared sensor. The flight process of moving targets in space, such as hypersonic glide vehicles (HGVs), is long, and the tracking of targets by satellites needs to cover the whole flight process of the targets. The whole tracking process requires the relay between multiple satellites. The cooperative scheduling of multiple satellites is a key

issue of current research. The continuous tracking of moving targets in space by multiple satellites is influenced by the target's maneuverability. The HGVs have gained considerable interest in recent years because of their high speed and high maneuverability. The coherence of satellite tracking can be greatly affected by their maneuver uncertainty. Therefore, this paper addresses the problem of cooperative tracking of HGVs by constellations.

*1.2. HGV Maneuver Intention Prediction*

HGVs typically use the boost-glide-dive flight mode, in which the vehicle is in the gliding phase during most of the flight time [2–4]. The gliding of HGVs is non-inertial, with higher flight speed, lower flight altitude, and stronger maneuverability than inertial trajectory targets [5].

Numerous scholars and laboratories have researched to solve the flight track prediction problem of HGVs. The main research directions include adaptive filtering algorithms [6–11] and interacting multiple model algorithms. For actual trajectories that are comparable to the model's trajectory, the single-model adaptive filtering algorithm has a high prediction accuracy [12]. However, it has a low prediction accuracy for trajectories with high uncertainty and different combinations of maneuver sequences. The interacting multiple model algorithm (IMM) [13] can infer similar models in the established model set based on the target observations and their associated computational quantities. Suitable models are selected for track prediction in different flight phases of the target. At present, the geographic location model and aerodynamic model of HGVs are well established, which can constitute the conventional maneuver model of HGVs [1,14,15]. In order to adapt the model set to all possibilities of the maneuver sequences, the number of models in the model set becomes increasingly large, and the complexity of the probability transfer matrix required for model selection increases. As a result, computational speed and dynamic performance are degraded. The probabilistic transfer matrix cannot accurately describe the uncertainties of the target maneuver strategy, so the interacting multiple model algorithm gradually becomes difficult to adapt to the increasingly complex maneuver form of HGVs. Therefore, in recent years, machine learning approaches have been increasingly adopted to solve the trajectory prediction problem for HGVs [16,17]. Herein, a convolutional neural network (CNN) and a long short-term memory network (LSTM) are combined into a classification network to classify the trajectories of HGVs. Then prediction networks such as LSTM [18], gated recurrent units (GRU), and bidirectional recurrent neural networks (Bi-RNNs) [19] are applied to the trajectory prediction problem of HGVs in the absence of a large set of models.

A satellite constellation tracking system relies on the calculations of the HGVs trajectory prediction algorithm for mission planning. Although the research on the trajectory prediction of HGVs has become increasingly mature, the uncertainty problem of maneuver sequences still has a large impact on trajectory prediction and tracking mission planning [20–22]. Therefore, the study of the uncertainty problem of maneuver intention sequences of HGVs has been gradually developed [23]. Theoretical trajectory generation of HGVs consists of design, optimization, and control processes, and each of its maneuvers is purposeful. Maneuver intentions can be divided into two categories according to the direction: longitudinal maneuver intentions and lateral maneuver intentions. These two types of maneuver intentions each have certain laws and are strongly coupled with each other.

A recurrent neural network (RNN) is a chain network structure consisting of connected recurrent units [24]. RNN is capable of handling time sequence information problems, but there is gradient disappearance and gradient explosion during the training process. Long short-term memory (LSTM) is a variant of the RNN with a special gate mechanism. It effectively solves the gradient disappearance problem that occurs during the training of traditional RNN for long time sequences [25]. The LSTM network provides feature extraction, memory functionality and high-dimensional information processing [26]. Moreover, the stacking layer mechanism can enhance the power of the LSTM to cope with more complex recognition and prediction problems of temporal correlation. In addition,

using multilayer LSTM networks to predict the target maneuver type is also a current maneuver intention prediction method with strong adaptability and high prediction accuracy [18]. The LSTM network can be used to approximate the dynamic relationships between aerodynamic parameters and displacements by aerodynamic analysis of HGVs at variable Mach numbers and mean angles of attack. In summary, the establishment of the maneuver intention model for HGVs is mainly based on maneuver intent type modeling. The consideration of longitudinal and lateral maneuver magnitudes mainly relies on the artificial classification of trajectories into several classes, which are then recognized by LSTM networks [27]. A multi-channel neural network is a parallel connection of multiple single-channel neural networks. Each channel extracts different patterns in the time sequences, and then the outputs of multiple channels are fused together to achieve the recognition and prediction of complex temporal relationships [19]. In this paper, in solving the recognition and prediction problem of maneuver intention sequences of HGVs, the number of stacked network layers is adjusted on the basis of multilayer LSTM networks to improve the recognition and prediction accuracy. Then, multiple LSTM networks are connected in parallel by using multi-channel neural networks. Each channel extracts the temporal association information of different time steps, and then the features of multiple time steps are fused to obtain rich trajectory features, and then the maneuver intention results are obtained by the fully connected network.

### 1.3. The Distributed Task Planning Algorithm

The mission planning algorithms of the satellite constellation tracking system primarily include centralized [28–30] and distributed planning [31,32]. All tasks in centralized task planning are scheduled by a main controller. The main controller needs to collect information from all satellites and provide planning solutions for them. The communication lag and delayed responsiveness in the real time limit centralized mission planning restrict its ability to adapt to the trajectories of highly maneuverable HGVs.

Distributed task planning algorithms include multi-agent task planning algorithms [33–36], auction-based planning algorithms [37], and consensus-based planning algorithms [38–40]. Among these, the consensus-based task planning algorithms have superior planning results and the fastest convergence [38]. The LCBBA is a mature and well-performing consensus algorithm [41]. The first two of these groups of methods require the selection of the information hub satellite in the constellation. Each satellite exchanges information with the information hub satellite to negotiate a planning solution. The consensus-based mission planning algorithm does not have an information hub and a main controller. In order to reach the global goal eventually, each satellite in the constellation needs to follow the same process, including self-evaluation, the satisfaction of local constraints, communication, and cooperation [42]. Considering the dynamic nature of the satellite's own state information and mission information in the actual mission, there are differences in the state information of other satellites collected by each satellite. Therefore, the LCBBA considers a local consensus based on the consensus-based bundle algorithm (CBBA) [41], which is suitable for the actual situation in satellite constellation tracking systems.

During the relay tracking of a target by two satellites, the reassigned satellite needs to choose a suitable time to continue tracking the target before the previous satellite ends its observation. The target's trajectory varies with time and has a high degree of uncertainty. A satellite's visibility of the target is affected by the uncertainty of the target trajectory and affects the selection of the handover satellite and handover time. Therefore, it is necessary to consider the uncertainty of the target trajectory in the mission decision method. Previously, a cooperative task scheduling algorithm to handle multiple autonomous satellite systems was proposed [35]. The algorithm was designed with a dynamic-distributed structure and a single-satellite scheduling algorithm that balances the robustness and stability of the system. The algorithm was also applied to two autonomous coordination mechanisms, the improved contact network protocol (ICNP) and the blackboard model (BM), and useful conclusions were obtained. In addition, adding the robustness parameter to the value

function is a way to improve the ability of the LCBBA to adapt to the target's uncertainty. In this paper, we consider the uncertainty of the maneuver intentions of HGVs and add an uncertainty factor to the value function of the satellite-independent decision. Sensor-switching strategies are then designed to accommodate the effects of target maneuvers on satellite visibility.

*1.4. Research Content*

In this paper, the research aims to solve the tracking problem of a satellite constellation tracking system for HGVs. This study's main research components and critical contributions are as follows: (a) A learnable maneuver intention dynamics model for HGVs was established. (b) A parallel-stacked LSTM neural network (PSLSTM) was designed to recognize and predict the maneuver sequence of the HGV. (c) An intention uncertainty factor is added to make the satellite consider the uncertainty of the target's maneuver intention when selecting the target. On the basis of that, a sensor-switching strategy is designed to guide the satellite in choosing the timing of the handover.

In the simulation analysis, the maneuver intention prediction capability of the PSLSTM network and two currently popular network structures: the multilayer LSTM (M-LSTM) and the dual-channel and bidirectional neural network (DCBNN) were tested for comparison. From the comparison results, it can be seen that PSLSTM recognizes the maneuver types of HGVs with more than 96% accuracy and has a higher accuracy of the maneuver intention prediction. Then SS-LCBBA was compared with three advanced distributed mission planning algorithms, LCBBA, BK and ICNP, in a satellite constellation tracking scenario. From the value function curve, SS-LCBBA improved the cumulative tracking score by 5–10% in the task of tracking HGV. It can be seen that SS-LCBBA has better tracking performance for HGVs in the multi-satellite collaborative scheduling problem.

The subsequent sections of this paper are organized as follows: In Section 2, a maneuver intention model for HGVs is established. In Section 3, a parallel-stacked LSTM neural network for maneuver intention prediction of HGVs is designed. In Section 4, the performances of three parallel forms of intention prediction networks are compared. Then, the SS-CBBA algorithm is compared to three baseline algorithms. In Section 5, the concluding part of this paper's research is discussed.

## 2. Establishing and Recognizing the Maneuver Intention of HGVs

This section first establishes the maneuver intention model. To solve the partial complexity problem of intention recognition and prediction, the maneuver intentions of the HGV are decomposed into four intention parameters with weak correlations. These results for each parameter have simple regularity. Finally, the parallel-stacked LSTM neural network (PSLSTM) is designed to recognize and predict the intention parameters.

*2.1. Establishing the Maneuver Intention Model*

The HGV at the $i$-th maneuver intention can be described as a combination of intent elements $A_1^i$, $A_2^i$, $A_3^i$, and $A_4^i$, where the superscript represents the number of maneuver intention sequences. Its ranges are from 0 to the entire flight of the HGV. The subscript represents the intention parameter sequence number.

2.1.1. The Longitudinal Intention Model

As shown in Figure 1, $T$ is the flight time and $H$ is the altitude of the HGV. The longitudinal maneuver intention types are divided into two categories, quasi-equilibrium gliding ($QEG$) and skip gliding ($SG$). The first maneuver intention parameter is used to describe the longitudinal maneuver intention type in the following form.

$$A_1^i = \begin{cases} 0, QEG \\ 1, SG \end{cases} \tag{1}$$

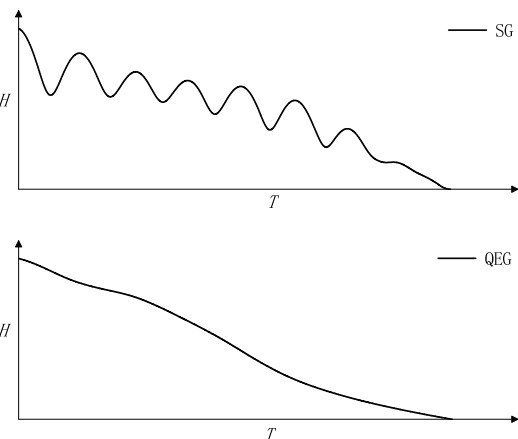

**Figure 1.** The schematic diagrams of *SG* and *QEG*.

Since the difference between *QEG* and *SG* trajectories is obvious, the $A_2^i$ intention parameter is calculated differently.

When an HGV's longitudinal maneuver is intended to be *QEG* ($A_1^i = 0$), $\dot{\gamma} \leq 0$ is maintained during the flight and the flight-path angle $\gamma$ is negative and monotonically non-incremental. Here $\dot{\gamma}$ is the rate of change of $\gamma$.

In this paper, the rate of change is defined as follows: the rate of change is the first order derivative of the variable $x$ with respect to time $t$, denoted as:

$$\dot{x} = \frac{\partial x}{\partial t} \tag{2}$$

The rate of change in altitude of HGV is described as:

$$\dot{h} = v * \sin(\gamma) \tag{3}$$

where $v$ is the velocity. As such, the relationship between $\dot{h} - v$ is determined by $\sin(\gamma)$. Therefore, $\sin(\gamma)$ is chosen as the characteristic parameter $A_2^i$ in the *QEG* maneuver intention. $A_2^i$ is denoted as:

$$A_2^i = \sin(\gamma) \tag{4}$$

When the longitudinal maneuver intention is *SG* ($A_1^i = 1$), the HGV repeatedly makes slide over movements. The flight-path angle is not monotonic. Therefore, ignoring the Earth's rotation, The variation of $\gamma$ with time is as follows:

$$\dot{\gamma} = \frac{L \cos \sigma}{mv} + \frac{v}{r} \cos \gamma - \frac{g}{v} \cos \gamma \tag{5}$$

where $L$ is the lift force, $g = 9.81 \text{ m/s}^2$ is the gravitational acceleration, $m$ is the mass of the vehicle, and $\sigma$ is the bank angle. $\dot{\gamma}$ is a visual representation of the longitudinal maneuver intention of HGVs. As seen from the above equation, the parameters mainly include the longitudinal component of the lift force $L \cos \sigma$, $v$, and $m$. As the parameters in the second and third terms of Equation (5) are all observable measurements, the first term $\frac{L \cos \sigma}{mv}$ is used in this paper as the characteristic parameter $A_2^i$ in the *SG* maneuver intention, which is written as:

$$A_2^i = \frac{L \cos(\sigma)}{mv} \tag{6}$$

2.1.2. The Lateral Intention Model

$\sigma$ is a visual description of the maneuver direction. The sign of $\sigma$ is opposite to the maneuver direction of the trajectory. Thus, the lateral maneuver intention parameter $A_3^i$ is written as:

$$A_3^i = \text{sign}(-\sigma)$$
$$= \begin{cases} -1, \sigma > 0 \\ 0, \sigma = 0 \\ 1, \sigma < 0 \end{cases} \tag{7}$$

The HGVs researched in this paper are unpowered gliding vehicles; hence, the lateral maneuver is realized by changing the HGV's attitude. Thus, $\sigma$ and the lateral component of the lift force on the vehicle is the visual representation of the lateral maneuver intention of the vehicle. As such, the lateral maneuver intention parameter $A_4^i$ is denoted as:

$$A_4^i = \frac{L \sin(\sigma)}{mv \cos(\gamma)} \tag{8}$$

where $L$, $\sigma$, and $m$ are unobservable measurements, and $v$ and $\gamma$ are observable measurements.

### 2.2. Parallel-Stacked LSTM Neural Network

The LSTM algorithm solves the long-term dependency problem of the traditional RNN algorithm. Furthermore, the gate setting also effectively solves the gradient disappearance and gradient explosion problems. The LSTM algorithm is suitable for learning and memorizing the temporal relationships in long sequence data. To enrich the fitting ability of LSTM, three LSTM network layers are stacked into a multilayer network using multilayer neural networks. A multilayer LSTM network connected in series is called a single-channel LSTM network. The maneuver intention sequences of HGVs have a short-time strong correlation. However, the LSTM learns and remembers all the data input to the network, including the sequence of maneuver intentions for the full time of HGVs. To force the network to focus on the short-term interdependence between data without losing its long-term memory capability, single-channel LSTM networks are connected in parallel using multi-channel neural networks. The network structure of PSLSTM is shown in Figure 2.

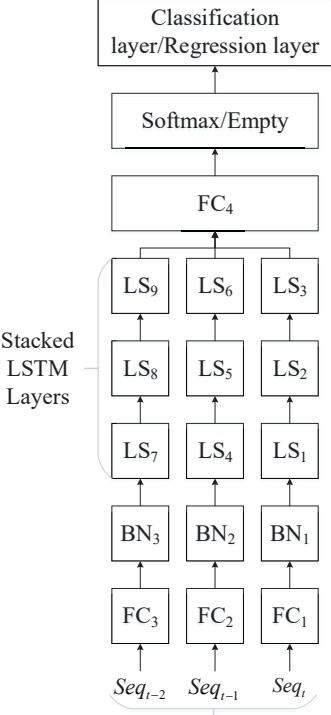

**Figure 2.** The parallel-stacked LSTM networks for intention recognition.

PSLSTM network structure can be described as follows: Sequence data are input through the sequence layer and then connected to the stacked LSTM network, where the number of parallel networks $P_n$ is a variable value. The sequence data are input in different ways for different $P_n$. When $P_n = 1$, the network's input is $Seq_t$, and the output is $R_t$. It is the stacked LSTM network. Additionally, when $P_n = 2$, the input of the network not only includes $Seq_t$, but $Seq_{t-1}$ is inputted as the auxiliary data. $Seq_{t-1}$ and $Seq_t$ are input to the parallel network in turn. These two sequence data are pooled to $FC_4$ after passing through different stacking networks, and the network results $R_t$ are output after $FC_4$ processing. When $P_n = 3$, the auxiliary data are $Seq_{t-2}$ and $Seq_{t-1}$, latter process is similar to $P_n = 2$.

The fully connected layer (FC) is represented as:

$$\begin{aligned} \iota &= \text{FC}(Seq), \\ &= \tanh(wSeq + b), \end{aligned} \tag{9}$$

where $w$ and $b$ are trainable parameters and $Seq$ is the input sequence.

The batch normalization layer is used to avoid the gradient disappearance problem and to reduce the probability of the overfitting phenomenon. With such a layer, the recognition network's generalization ability can also be improved. The mathematical description of the batch normalization layer is as follows:

$$\begin{aligned} \iota_{bn} &= BN(\iota) \\ &= \omega_{bn} \frac{\iota - \bar{\iota}}{\sqrt{\sigma_\iota^2 + \varepsilon}} + b_{bn}, \end{aligned} \tag{10}$$

where $\iota$ is the input of the batch normalization layer; $\bar{\iota}$ and $\sigma_\iota$ are the mean and standard deviation of $\iota$, respectively; $\omega_{bn}$ is the output of the batch normalization layer.

Compared with traditional neural networks, in addition to the hidden layer state $\iota_n$, LSTM networks maintain a cell form $C_n$ for storing temporal association relations. The network updates the cytosolic state $C_k$ through the forgetting and input gates. The network updates the cytosolic state $C_k$ through the forgetting and input gates. The cytosolic state $C_n$, the previous hidden layer state $\iota_{n-1}$, and the current input $Seq_n$ are integrated into the output gate to obtain the network output. With such a cell form, the network output can realize the recent temporal and long-term temporal relationship between memory and inference. In this paper, the LSTM layer is described as:

$$[\iota_n, C_n] = LS(\iota_{n-1}, C_{n-1}, Seq_n) \tag{11}$$

where $\iota_{n-1}$ and $C_n$ are the previous hidden state and the cell state at the last moment, respectively; $Seq_n$ is the current input sequence; and $\iota_n$ and $C_n$ are the hidden state output and the cell state output, respectively.

The PSLSTM designed in this section is as follows:

$$\begin{cases} \iota_{n,13,5} = FC_{1\text{-}3}(Seq_n), \\ \iota_{n,2,4,6} = BN_{1\text{-}3}(\iota_{n,13,5}), \\ [\iota_{n,L1-L9}, M_{n,L1\text{-}L9}] = LS_{1\text{-}9}(\iota_{n-1,L1-L9}, M_{n-1,L1-L9}, \iota_{n,1,L1,L2,2,L3,L4,3,L5,L6,4,L7,L8}), \\ \iota_{n,7} = FC(\iota_{n,L9}), \\ P(A_k^{1:N} \big| Seq_{(1:N)}) = S_f(\iota_{n,7}), k = 1,3 \\ A_k^{1:N} = Cl\Big(P(L \big| Seq_{(1:N)})\Big), k = 1,3 \\ A_k^{1:N} = \text{Re}(\iota_{n,7}), k = 2,4 \\ n = 1, 2, \cdots, N \end{cases} \tag{12}$$

where $S_f(x)$ is a fully connected layer with the softmax function.

After the independent recognition of the above network, the maneuver intention sequence of the HGV can be obtained. The intention sequence and observations are

integrated and put into the joint maneuver intention prediction neural network. This network incorporates the learning of the four intention parameters and the correlation among the observations to predict the subsequent maneuver intention of the HGV.

The structure of the joint prediction algorithm using the above network is shown in Figure 3. To obtain the final maneuver intention prediction results, firstly, the observation information is preprocessed to fit the requirements of the maneuver intention recognition network. Then, the intention labels and intention parameters of HGVs are recognized based on the PSLSTM neural network designed above. Finally, the observation information and the maneuver intention recognition information are integrated and input into the joint maneuver intention prediction network.

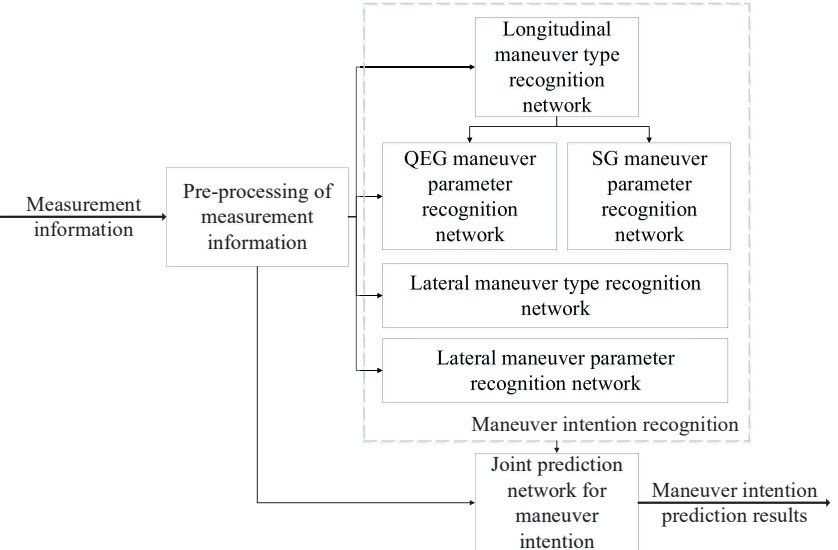

**Figure 3.** The joint prediction algorithm based on PSLSTM.

Among them, the observation information preprocessing process is shown in Figure 4. The observation information is shown below: $h$ is the altitude of the HGV. $\theta$ and $\phi$ are the latitude and longitude of the vehicle. $R_e$ is the radius of the Earth. $v$ can be expressed as:

$$v = \sqrt{\dot{h}^2 + R_e^2\,\dot{\theta}^2\cos^2\phi + R_e^2\,\dot{\phi}^2}$$ (13)

$\gamma$ can be expressed as:

$$\gamma = acos\left(\frac{\dot{h}}{\sqrt{R_e^2\dot{\theta}^2\cos^2\phi + R_e^2\dot{\phi}^2}}\right)$$ (14)

The rate of the heading angle $\dot{\psi}$ can be calculated by:

$$\dot{\psi} = acos\left(\frac{v * \dot{v} * \cos\gamma}{\sqrt{\dot{v}^2 + v^2}}\right)$$ (15)

Four tags $A_1$, $A_2$, $A_3$ and $A_4$ need to be recognized to determine the vehicle's maneuver intention. The input of the $A_1$ recognition network includes six parameters: $h$, $\dot{h}$, $v$, $\dot{v}$, $\gamma$ and $\dot{\gamma}$. The output is $A_1$. The $A_2$ recognition network is divided into the *QEG* and *SG* maneuver parameter recognition networks. The inputs of the $A_2$ recognition network are $h$, $\dot{h}$, $v$, $\dot{v}$, $\gamma$ and $\dot{\gamma}$. The output is $A_2$. The input of the $A_3$ recognition network includes five parameters: $\theta$, $\phi$, $\dot{\theta}$, $\dot{\phi}$ and $\dot{\psi}$. The output is $A_3$. The input of the $A_4$ recognition network is the same as the $A_3$ recognition network. Then the output is $A_4$.

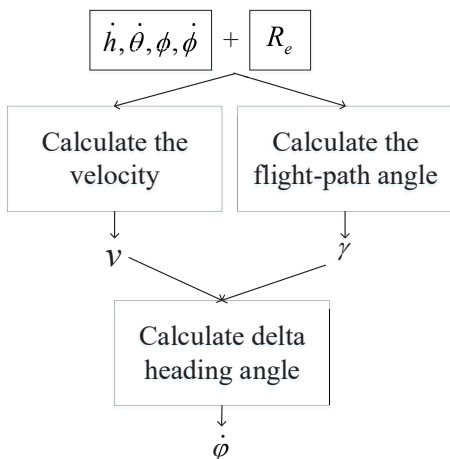

**Figure 4.** Pre-processing of measurement information.

The uncertainty of the future intention trend of the vehicle is intense, and the number of coupling relationships among the existing parameters is large. In the maneuver intention prediction stage, 15 parameters are put into the joint prediction network. The parameters include: $h, v, \gamma, \theta, \phi, \dot{h}, \dot{v}, \dot{\gamma}, \dot{\theta}, \dot{\phi}, \dot{\psi}$, and four intention parameters, $A_1, A_2, A_3, A_4$. Two fully connected layers are added to the network to extract the coupling relationship between the parameters. Then, a parallel-stacked LSTM neural network is added to learn and predict the temporal association between the data. Finally, a fully connected layer is added to map the learned data into the output form of four neurons to make a sequential output.

The intention prediction of HGVs has high complexity. It is nearly impossible to learn the laws of all maneuver intentions in a limited number of observations. Hence, the prediction network input data are expanded and enriched by the work carried out in the first two sections. In the observation data expansion and recognition, this method creates an independent recognition network for each maneuver intention. This design can make the fuzzy association relationship between data more precise in the form of intention parameters.

*2.3. Network Training*

2.3.1. Loss Function

The mean square error (MSE) was chosen as the loss function of PSLSTM, which is calculated as follows:

$$Loss = \frac{1}{k} \sum_{i=1}^{k} \left( \hat{y}_{T+1}^{i} - \hat{y}_{T+1}^{i} \right)^2 \tag{16}$$

where $k$ is the dimension of the network input and $\hat{y}_{T+1}^{i}$ is the output of PSLSTM. $y_{T+1}^{i}$ is the actual value of the data at the next time stamp. The MSE loss function amplifies the error between the prediction results and the actual data with a squared relationship, making the PSLSTM network more sensitive to errors.

2.3.2. Optimization Strategy

Adaptive moment estimation (Adam) was chosen as the optimizer for the training method. Adam is computationally efficient and fast in gradient descent, which is suitable for large-scale data and parameter training scenarios.

2.3.3. Training Strategy

To improve the recognition and prediction performance while considering the real-time computational constraints, we divide the model training into two parts: pre-trained and retraining. During pre-trained, the PSLSTM network is optimized by iteratively updating with the training data using the loss function and optimizer described above. The main purpose of pre-trained is to converge the validation error to a smaller value.

The pre-trained process takes a lot of time and computational resources to obtain the trained network parameters. The retraining process is executed before the actual prediction. First, we load the pre-trained PSLSTM network and then input the latest observed HGV data into the network for training. The number of retraining steps is much smaller than the number of pre-trained steps due to the computation time limitation, but this process still greatly enhances the performance of network recognition and prediction.

### 3. Local Consensus-Based Bundle Algorithm Method Considering Sensor Switching

In this paper, considering the maneuver intention uncertainty of HGVs, the loss coefficient is added to the value function of SS-LCBBA. Subsequently, a sensor-switching strategy is designed. SS-LCBBA adapts to the time-varying task sequences in satellite constellation tracking observation missions and improves the cumulative tracking score of tracking missions.

#### 3.1. Value Loss Factor

The traditional HGV altitude prediction is the center of the longitudinal uncertainty region (LUR). $Lr$ is the LUR range, which is described in the following form:

$$Lr = \begin{cases} [h - R_n * A_2 * v, h + R_n * A_2 * v], A_1 = 0 \\ [h - R_n(A_2 * v - g), h + R_n(A_2 * v - g)], A_1 = 1 \end{cases} \tag{17}$$

where $A_1$ and $A_2$ are output values of joint prediction networks, and $h$ and $v$ are obtained by the trajectory prediction algorithm. The LUR width coefficient $R_n$ and gravitational acceleration $g$ are constant values. The LUR of the vehicle is calculated separately according to longitudinal maneuver intentions $A_1$. Then, the widths of LUR coefficients are set independently according to different maneuver types.

$Tr$ is the lateral uncertainty region (HUR) range, which is described in the following form:

$$Tr_o^{\max} = \alpha_o + k_o |A_4| \cos\left(P - \frac{\pi}{2} * A_3\right) / (R_e + H) \tag{18}$$

$$Tr_o^{\min} = \alpha_o - k_o |A_4| \cos\left(P - \frac{\pi}{2} * A_3\right) / (R_e + H) \tag{19}$$

$$Tr_a^{\max} = \alpha_a + k_a |A_4| \sin\left(P - \frac{\pi}{2} * A_3\right) / (R_e + H) \tag{20}$$

$$Tr_a^{\min} = \alpha_a - k_a |A_4| \sin\left(P - \frac{\pi}{2} * A_3\right) / (R_e + H) \tag{21}$$

where $\alpha_o$ and $\alpha_a$ are the predicted values of the vehicle's latitude and longitude in the conventional trajectory prediction algorithm; $P$ and $H$ are the predicted values of the bank angle and the altitude; and $k_o$ and $k_a$ are the latitude and longitude HUR width coefficients.

As described in the literature, the LCBBA is an iterative two-phase algorithm. These two phases are a bundle building phase and a task consensus phase. In the bundle building phase, each vehicle greedily generates an ordered list of assignments. In the task consensus phase, conflicting duties are identified and resolved through local communication between neighboring satellites. The uncertainty loss factor and sensor rotation decision are improved for the bundle building phase, and the task consensus phase still uses the method of the LCBBA, which is not repeated in this paper.

The line of sight angle $\alpha_s$ of the satellite to the vehicle is defined as:

$$\alpha_s = \alpha_t - \alpha_e \tag{22}$$

where $\alpha_t$ is the line of sight angle of the satellite to the vehicle, $\alpha_e$ is the lower limit of the proximity observation angle and the line of sight angle of the satellite to the tangent of the Earth. The loss coefficients $L_l^n$ are described as follows:

$$L_l^n = \begin{cases} 1 - e^{-\frac{\alpha_s^n}{C_l \alpha_e^n}} & , 0 \leq \alpha_s^n - \alpha_e^n \leq \alpha_j \\ 0 & , \alpha_s^n - \alpha_e^n < 0 \\ 1 & , \alpha_s^n - \alpha_e^n > \alpha_j \end{cases} \tag{23}$$

The angular range $\alpha_j$ of the Earth's limbic interference and the scaling factor $C_l$ are constants.

The infrared sensor has a maximum detection distance constraint. The line of sight distance $D_s$ between the satellite and the vehicle has a detection accuracy degradation problem when approaching the sensor detection distance maximum $D_l$. The distance loss coefficient $L_t^n$ is described as follows:

$$L_t^n = \begin{cases} 1 & , D_s^n < D_a \\ 1 - e^{-\frac{D_s^n - D_a^n}{C_t(D_l - D_a^n)}} & , D_a^n \leq D_s^n \leq D_l \\ 0 & , D_s^n > D_l \end{cases} \tag{24}$$

where $D_a$ is the distance when tracking accuracy starts to decline. The constant $C_t$ is the decline scaling factor.

The observation angle loss coefficient $L_t^n$ and the distance loss coefficient $L_t^n$ are simultaneously used on the value function to obtain the cumulative tracking score in the following form:

$$C_{n,k} = L_l^n * L_t^n * C_{n,k}^R \tag{25}$$

where $C_{n,k}^R$ is the theoretical value of the value function. The joint action of two-loss coefficients makes the cumulative tracking score more adaptive to the uncertainty of the maneuver intention.

### 3.2. Sensor-Switching Strategy

When a satellite bids on a set of tasks with temporal correlation, the time it takes for infrared sensors to switch between HGVs must be considered.

The minimum time $T_{min}$ required to switch between tasks is calculated first. Then, the UR width $L_w$ considering the maximum rotation time $T_r$ needed for the UR of the vehicle's intention is determined. The best switching window for the satellite is selected. Sensors can rotate to the tracking position before the UR becomes wider.

As shown in Figure 5, the trajectory of the HGV can be divided into the maneuver period time (red area) and the maneuver gap period time (yellow area). According to the sequence of maneuver intentions of the vehicle, the widths of URs are compared and the interval with a smaller width is considered as the maneuver gap period time. The maneuver gap period time is a special case when the vehicle continuously maneuvers. The tracking accuracy of the satellite $Sat_n$ is constrained by the sensor's performance, the observation angle, and the distance. The green segment in the figure, "accuracy down" is a tracking accuracy degradation problem at the end of the visible time window. The bidding satellites need to consider the timing arrangement of the new task beam. The minimum time $T_f$ required for sensor back sweep is then calculated. The optimal switching window $[T_s^s, T_s^e]$ is calculated as shown in Algorithm 1:

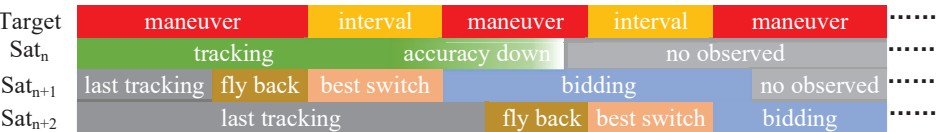

**Figure 5.** Sensor switching decision process.

---

**Algorithm 1** Sensor-switching strategy.

---

1: **if** $T_a \leq T_i^s$
2:     **if** $T_i^s \leq T_d^s$
3:         $T_s^s = T_i^s$
4:         $T_s^e = T_d^s$
5:     **else if** $T_i^e > T_d^s$
6:         $T_s^s = Max[T_d^s, T_a]$
7:         $T_s^e = Min[T_d^e, T_i^e]$
8:     **end**
9: **else if**
10:     $T_s^s = T_a$
11:     **if** $T_a < T_d^s$
12:         $T_s^e = T_d^s$
13:     **else**
14:         $T_s^e = Min[T_d^e, T_i^e]$
15:     **end**
16: **else if** $T_a > T_i^e$
17:     The optimal switch window is empty;
18: **end**

---

where $[T_i^s, T_i^e]$ is the maneuver gap time of the vehicle, $[T_d^s, T_d^e]$ is the tracking sensor accuracy degradation period time, and $T_a$ is the earliest arrival time.

The computational complexity of the task consensus phase in the SS-LCBBA algorithm is $O\left(\frac{(n^2\hat{m})^2}{2}\right)$, $\hat{m} = \max\limits_{i=1}^{n} \max\limits_{g=1}^{n} (m_{ig})$, where $n$ is the number of satellites. $m_{ig}$ is the number of communication time windows between two satellites. $\hat{m}$ indicates the maximum number of communication time windows between satellites during the mission. The computational complexity of the independent decision algorithm for each satellite is $O(u^2 + (u - 1) \cdot N \cdot d^2)$, where $u$ is the number of HGVs, $N$ is the length of the neural network input sequence, and $d$ is the dimension of each element in the input sequence. Hence, the computational complexity of the SS-LCBBA is $O\left(n \cdot \frac{(n^2\hat{m})^2}{2} \cdot (u^2 + N \cdot d^2 \cdot (u - 1))\right)$.

## 4. Simulation and Analysis

In this section, the performances of the intention recognition network and the joint prediction network are analyzed through a series of typical HGVs motion scenarios. The joint prediction network is compared with multilayer LSTM networks (M-LSTM) and dual-channel and bidirectional neural network (DCBNN). Compare the performance of several networks in coping with the HGV maneuver intention uncertainty problem. Subsequently, three distinct trajectories of HGVs with different maneuver forms for comparative simulations of baseline LCBBA, improved contract network (ICNP), blackboard model (BM) and SS-LCBBA are used to compare the tracking capabilities of the four algorithms for different maneuver sequence HGVs.

### 4.1. Simulation and Analysis of Networks

Seven recognition neural networks were trained on four intention parameters and the joint prediction network using flight trajectories of HGVs with different maneuver sequences. After that, the accuracies of three parallel forms of each neural network were compared. The maneuver intention recognition and prediction neural network is implemented with the PyTorch deep learning framework. The simulation platform is PyCharm. The simulation language is Python. The training was performed on a computer equipped with an i7-8700 processor, 16 GB of RAM, and an NVIDIA GTX 1060 graphics card.

### 4.1.1. Architecture

The structure of the intention recognition network is shown in Figure 2, where the number of neurons in each fully connected layer of the stacked network was set to 128; the three LSTM layers were formed as 512, 256, and 128. The *tanh* activation function was chosen for the state activation function of the intent recognition network. The weight matrix was orthogonally initialized to avoid gradient disappearance or gradient explosion at the beginning of the training equation. Then, we set the initial value of the bias to zero. The length of the sequence was set to 200.

### 4.1.2. Training Data

The detailed training data are shown in Appendix A. According to the aerodynamic parameters and maneuver forms of HGVs, 12 initial conditions (in Table A1), 14 longitudinal maneuver sequences (in Table A2), and 40 lateral maneuver sequences (in Table A3) were set. In total, 6720 flight trajectories are combined, including 3360 *QEG* maneuvers and 3360 *SG* maneuvers. Among them, 5600 training data trajectories, 560 validation data trajectories, and 560 test data trajectories were selected randomly.

Since the units of different dimensions of the training data may be different, the training data need to be dimensionless. This processing is to be executed before training. In this study, all training data were normalized using the min-max scaling method as follows:

$$X_{norm} = \frac{X - X_{\min}}{X_{\max} - X_{\min}} \tag{26}$$

where $X$ represents the training data; $X_{\min}$ and $X_{\max}$ are the minimum and maximum values in the training data at that step.

### 4.1.3. Training Parameters

We used the Adam solver with a batch size of 64 and a training period of 400. The cross-entropy loss function was used, and training could be ended earlier if the loss values converged. The learning rate was set to 0.005, the learning rate decline period was 100, and the learning rate decline factor was 0.2.

### 4.1.4. Comparison of Training Loss of Different Networks

Figure 6 shows the loss change curve during PSLSTM model training. Equation (16) is used to calculate the training loss. The number of training epochs is represented by the lateral coordinate, and the average loss within each epoch is represented by the vertical coordinate.

As shown in Figure 6a, the A2 recognition network has the fastest decreasing loss curve. Because after the A1 recognition network divides the longitudinal maneuver types of HGVs into two types, *SG* and *QEG*, the A2 recognition network of each type has a stronger variation pattern with time and can be visually described by the altitude and the velocity of HGV. Because the intention sequence of lateral maneuvers of HGVs is more complex and random, the A3 and A4 recognition networks have a slightly slower decreasing speed than the other two networks.

In this paper, the multilayer LSTM (M-LSTM) and the dual-channel and bidirectional neural network (DCBNN) are chosen as the comparison models for the joint prediction network. M-LSTM architecture consists of a fully connected layer, two LSTM layers, and a softmax layer. The M-LSTM model employs the hierarchical principle to recognize and predict multiple parameters. DCBNN is a two-channel structure with a fully connected layer and a bidirectional gated recurrent unit in each channel (Bi-GRU). To compare the capabilities of the networks, the number of neurons in each LSTM layer and GRU layer is set to 128, and the training parameters of the network remain the same as in Section 4.1.3. For specific details of the M-LSTM and DCBNN algorithms, please refer to [18,19].

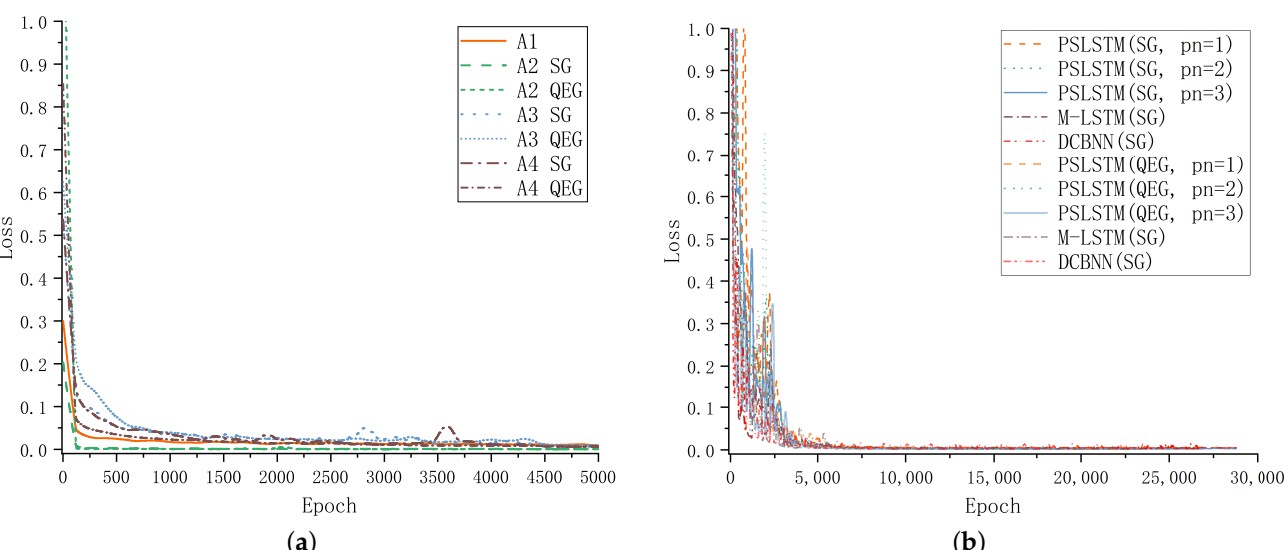

**Figure 6.** Network Training Loss: (**a**) is recognize network training loss and (**b**) is prediction network training loss.

As shown in Figure 6b, the training loss variation curves of the joint prediction network with M-LSTM and the DCBNN. The training loss of the PSLSTM network decreases slightly slower than that of M-LSTM and DCBNN because PSLSTM has more network layers and neurons. The prediction network of the *SG* trajectory declines slower than that of the *QEG* trajectory and has a relatively higher convergence value. Because the trajectory variation of *SG* trajectory is more complex, the network needs more iterations to learn the complex maneuver sequence. The training loss of the PSLSTM network converges to 0.0479 for the intention prediction network of *SG* trajectory, and 0.0395 for the intention prediction network of *QEG* trajectory. Hence, the training loss convergence value of the PSLSTM network is lower than that of M-LSTM and DCBNN.

### 4.1.5. Results

Networks $A_1$ and $A_3$ are intention label classification networks; thus, the accuracy rate was used as an evaluation condition to measure the effectiveness of the network in the following form:

$$P = \frac{\sum_{n=T_s}^{n=T_e} \sigma_n}{T_e - T_s} \tag{27}$$

where $\sigma_n$ is the success mark, with 1 representing successful and 0 representing unsuccessful. $T_s$ and $T_e$ are the start time and end time, respectively.

The other two intention networks, $A_2$ and $A_4$, are regressor networks, and the root mean square error (RMSE) is used to measure the accuracy of parameter regression. For specific details of the RMSE, please refer to [9,43]. The specific form is as follows:

$$R = \sqrt{\frac{1}{T_e - T_s} \sum_{n=T_s}^{n=T_e} \left( y_p^n - y_r^n \right)^2} \tag{28}$$

where $y_p^n$ and $y_r^n$ are the parameter regression values and the parameter real values, respectively.

The test comparison results of the intention network are explained below. Based on the same training and test data, the accuracy of the stacked networks with the numbers of parallel networks $P_n = 1, 2, 3$ are compared, and the average, minimum, middle, and maximum values of the accuracy calculation results act as the basis for comparison.

Case 1: $A_1$ recognition network

$A_1$ marks the longitudinal maneuver form of the HGV in the process of determining the maneuver intention of the HGV. The input of the $A_1$ recognition network includes six parameters: $h$, $\dot{h}$, $v$, $\dot{v}$, $\gamma$, and $\dot{\gamma}$. The output of the $A_1$ recognition network is divided into two labels, *QEG* and *SG*.

As shown in Figure 7, since the longitudinal maneuver intention of the HGV has apparent patterns in the parameter representation, the recognition accuracy is approximately 100% for the parallel network numbers $P_n = 1$ and 2. Therefore, it ensures high learning efficiency and prevents the overfitting phenomenon at the same time.

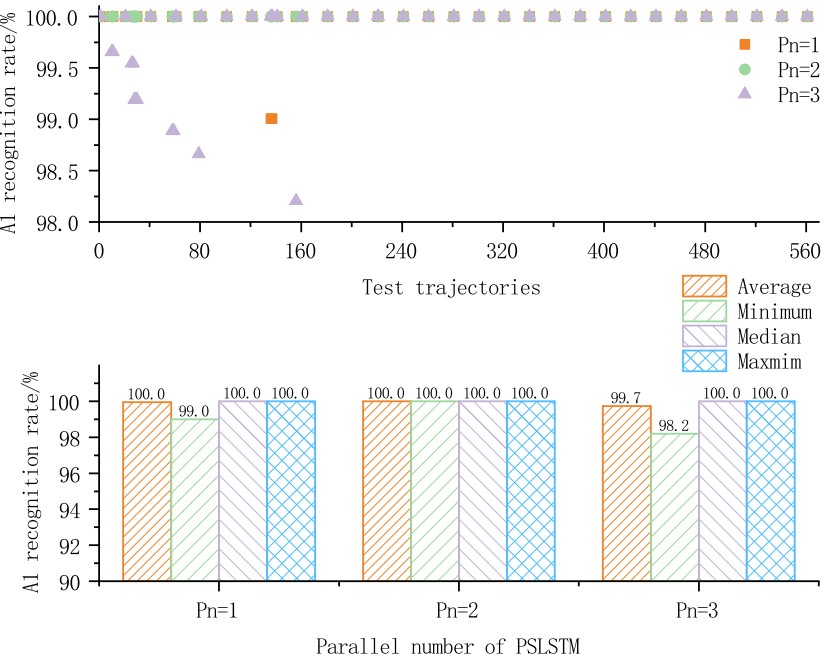

**Figure 7.** A1 recognition rate.

Case 2: $A_2$ Recognition Network

Section 2.1 shows that there are two different laws for $A_2$ when $A_1$ is 0 (*SG*) or 1 (*QEG*). The $A_2$ recognition network is divided into $A_2(SG)$ and $A_2(QEG)$ recognition networks. Since $A_2$ is also a parameter characterizing the longitudinal maneuver intention of the HGV, the input of the $A_2$ recognition network is the same as that for $A_1$. The parameter complexity and temporal correlation of $A_2$ are higher than those of $A_1$. The longitudinal dynamics of the HGV of *SG* trajectory changes into an oscillating trend. The root means the square error of the network recognition result is the smallest, as shown in Figure 8a. As shown in Figure 8b, the network with $P_n = 2$ has the best recognition accuracy for the vehicle of *QEG* gliding, as the longitudinal dynamics of the vehicle change gently and regularly.

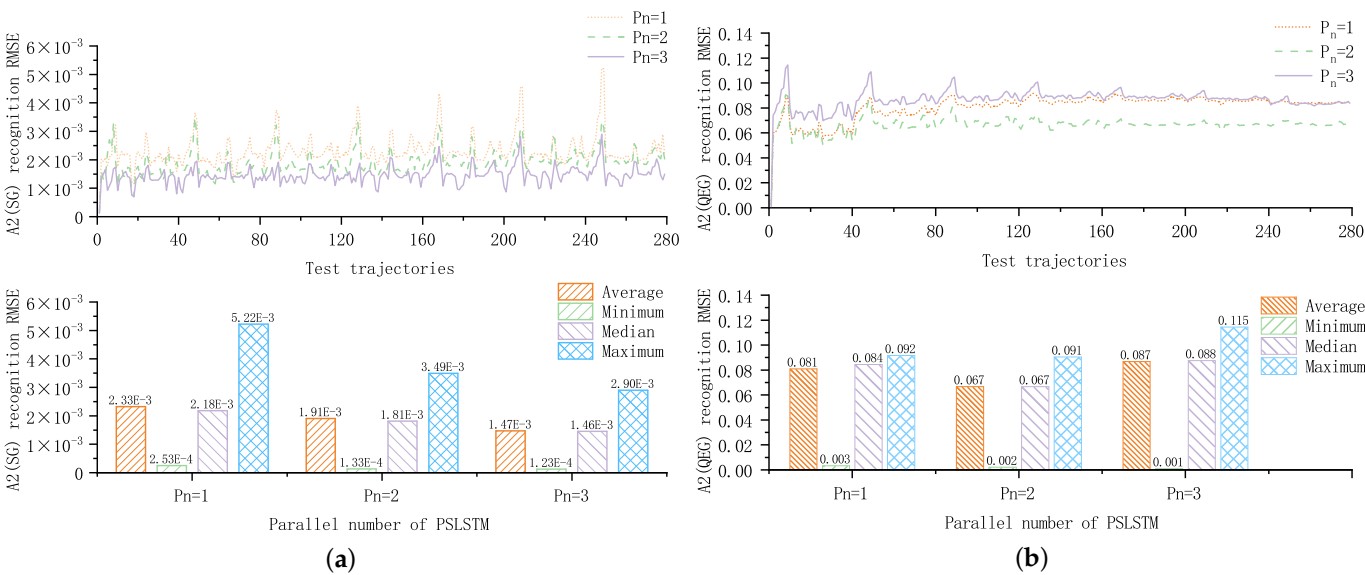

**Figure 8.** Recognition error of A2: (**a**) is the A2 (*SG*) recognition error curve and (**b**) is the A2 (*QEG*) recognition error curve.

Case 3: $A_3$ Recognition Network

The lateral maneuver intention label $A_3$ is a classification recognition label to recognize the lateral maneuver direction of the HGV. The input of the $A_3$ recognition network includes five parameters: $\theta$, $\phi$, $\dot{\theta}$, $\dot{\phi}$ and $\dot{\psi}$. The output of the recognition network with $A_3$ is classified into three categories: left maneuver, right maneuver, and no lateral maneuver.

Since the laws of the observed parameters differ greatly when the longitudinal intention types are different, the prediction network for the lateral maneuver intention $A_3$ still needs to be divided into *SG* and *QEG*. As shown in Figure 9a,b, although the recognition results of the three network forms are close, it can still be seen that the minimum value of the network with $P_n = 3$ evaluation results is greater.

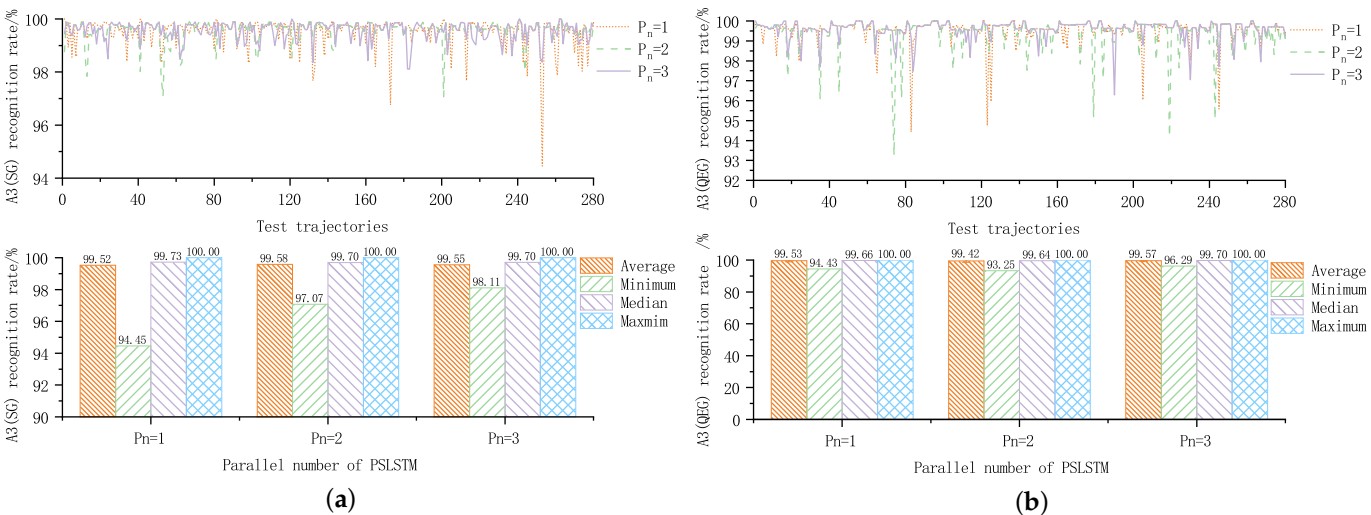

**Figure 9.** Recognition error of A3: (**a**) is the A3 (*SG*) recognition error curve and (**b**) is the A3 (*QEG*) recognition error curve.

Case 4: $A_4$ Recognition Network

The lateral maneuver intention parameter $A_4$ characterizes the HGV lateral force. Since the lateral maneuver of the HGVs relies on the adjustment of bank angle, the onboard infrared sensor cannot precisely recognize the change of the bank angle. Hence, the network input still uses five parameters related to the lateral maneuver intention ($\theta$, $\phi$, $\dot{\theta}$, $\dot{\phi}$, $\dot{\psi}$).

Moreover, the lateral maneuver label $A_3$ is added as the reference value for the lateral maneuver direction. From Figure 10a, it can be seen that, compared with $P_n = 1$ or 2, the $P_n = 3$ network got the highest recognition accuracy for *SG* trajectories. The mean, median, and minimum values of the root-mean square error of the recognition results are relatively smooth. However, the maximum value is larger because the correlation law between the input parameters and the magnitude of the lateral force is not obvious enough. From Figure 10b, it can be seen that for *QEG* trajectories, the maximum, median, and minimum values of the network with $P_n = 2$ are slightly smaller than those of the network with $P_n = 1$, but the average values are similar.

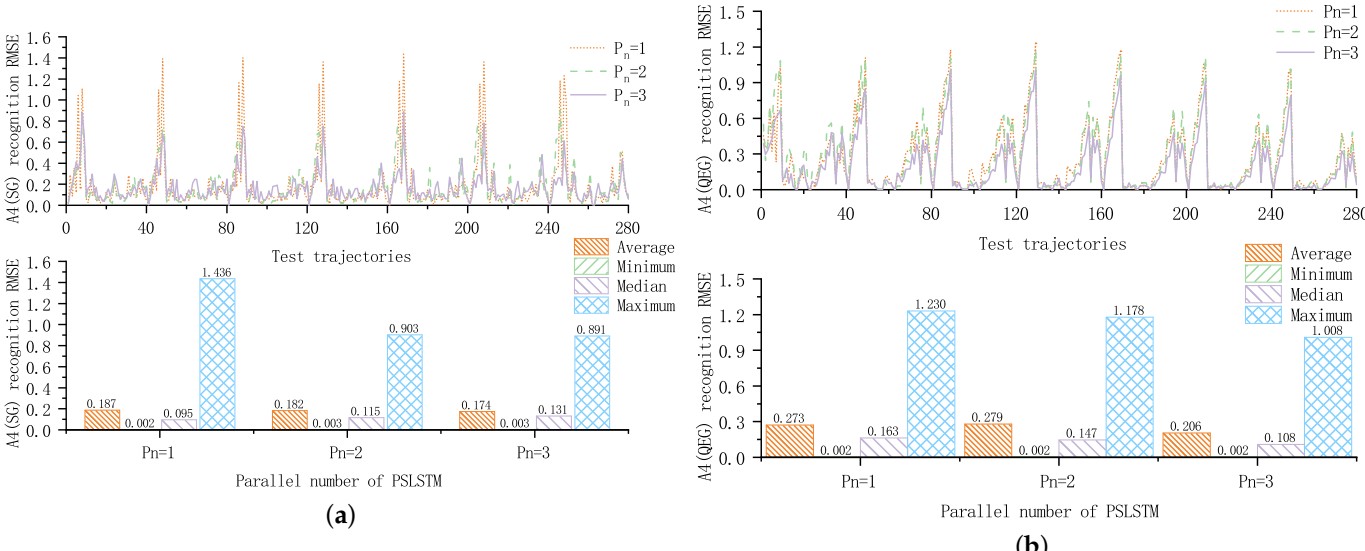

**Figure 10.** Recognition error of A4: (**a**) is the A4 (*SG*) recognition error curve and (**b**) is the A4 (*QEG*) recognition error curve.

The experiment results of the above intention parameter recognition network show that the stacked LSTM network can recognize the maneuver intention of the HGVs by observations, and the network with $P_n = 2, 3$ improves the success rate of maneuver intention recognition and accuracy compared to the network with $P_n = 1$. For the recognition of longitudinal maneuver type $A_1$, all three parallel forms can achieve a relatively high success rate. For the longitudinal maneuver intentions, the short-time correlation of the *SG* trajectory is stronger, whereas the short-time variation of the *QEG* trajectory is more flat. The network with $P_n = 3$ is more focused on the correlation between short time data, so it is more suitable to be used to recognize the longitudinal maneuver intention of *SG* trajectory. In contrast, the altitude change of the *QEG* trajectory is gentler. The observed data change less with time in instances of lateral maneuvers; hence, the network with $P_n = 2$ can achieve better recognition results.

Case 5: The Joint Prediction Network for Maneuver Intentions of *SG* Trajectories

The intention recognition network was used as a data-providing network for the joint prediction network. First, 10% of each trajectory data was used as the observation data. As described in Section 2.3.3, the pre-trained network was loaded and the observed data were used as retraining data to retrain the model in several steps. The intention parameters of the retraining data were recognized by the intention prediction network. The prediction simulation was performed for the HGVs' subsequent intention sequence to compare the prediction accuracy of the intention parameters of the PSLSTM with $P_n = 1, 2, 3$, M-LSTM, and DCBNN. Where M-LSTM and DCBNN have been described in Section 4.1.4. Each prediction required an input sequence length of 50. Since the output length of the intention recognition network was 200, it was divided into four segments, each with a length of 50. The prediction network divided the sequence output into four segments, each with a length

of 50. Then, these segments and the processed observations were arranged and input into the joint prediction network. The comparison results are as follows:

As shown in Figure 11, the simulation results of the maneuver intention prediction network for *SG* trajectory show that the prediction accuracy of the parallel-stacked neural network for the intention parameters is generally higher than that of the series stacked network. Additionally, the parallel network with $P_n = 3$ has higher prediction accuracy for the *SG* trajectory intention parameters $A_1$, $A_2$, and $A_3$. The three PSLSTM networks have similar prediction accuracy for the intention $A_4$, although the average value of the error at $P_n = 2$ is slightly smaller than those of the other two. M-LSTM and DDCNN also have the ability to maneuver intention prediction, but the prediction accuracy is lower than that of PSLSTM. DCBNN has the advantage of dual-channel and bidirectional recurrent network units, and the prediction accuracy is slightly higher than that of M-LSTM.

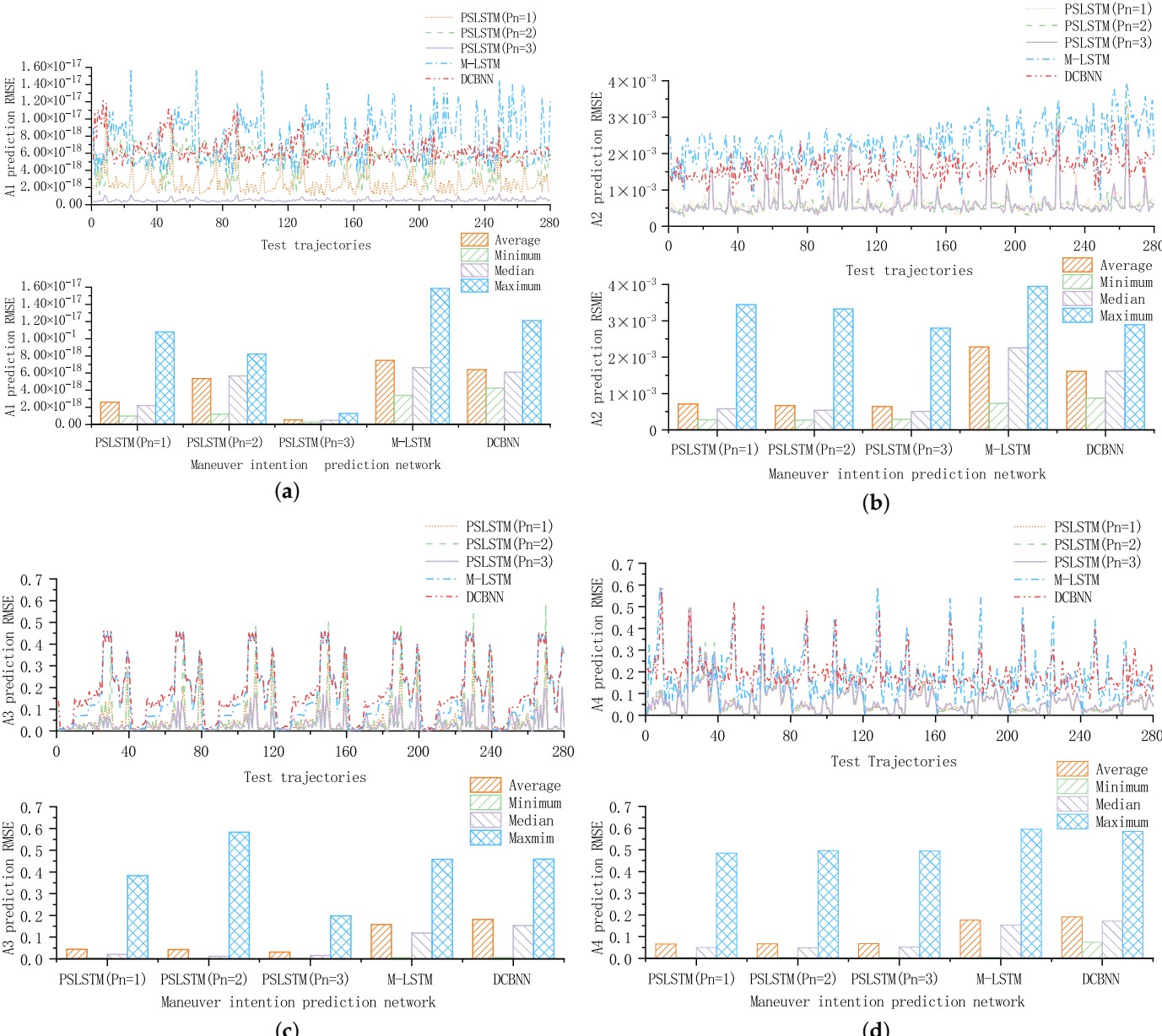

**Figure 11.** Prediction error of the maneuver intention prediction networks (*SG*): (**a**) is the error of A1 prediction, (**b**) is the error of A2 prediction, (**c**) is the error of A3 prediction, and (**d**) is the error of A4 prediction.

Case 6: The Joint Prediction Network for Maneuver Intentions of *QEG* Trajectories

The following is a comparison of the networks with $P_n = 1, 2, 3$ in parallel form:

The longitudinal flight path of the *QEG* trajectory is smoother than the *SG* trajectory. In contrast, the maneuver intention prediction network is more focused on learning the patterns of the lateral maneuver sequences. As such, the accuracy of the joint prediction results of the *QEG* trajectory is generally higher than that of the *SG* trajectory. As shown in Figure 12, it can be found that the prediction accuracies of the intention parameters $A_1 - A_3$ are better when the network parallel number $P_n = 2$ of PSLSTM. However, there was no significant difference in the prediction accuracies of the three PSLSTM for $A_4$. The difference in prediction accuracy between M-LSTM and DCBNN is smaller, and DCBNN has an advantage in the prediction of $A_2$ and $A_4$, but the accuracy is still lower than that of PSLSTM.

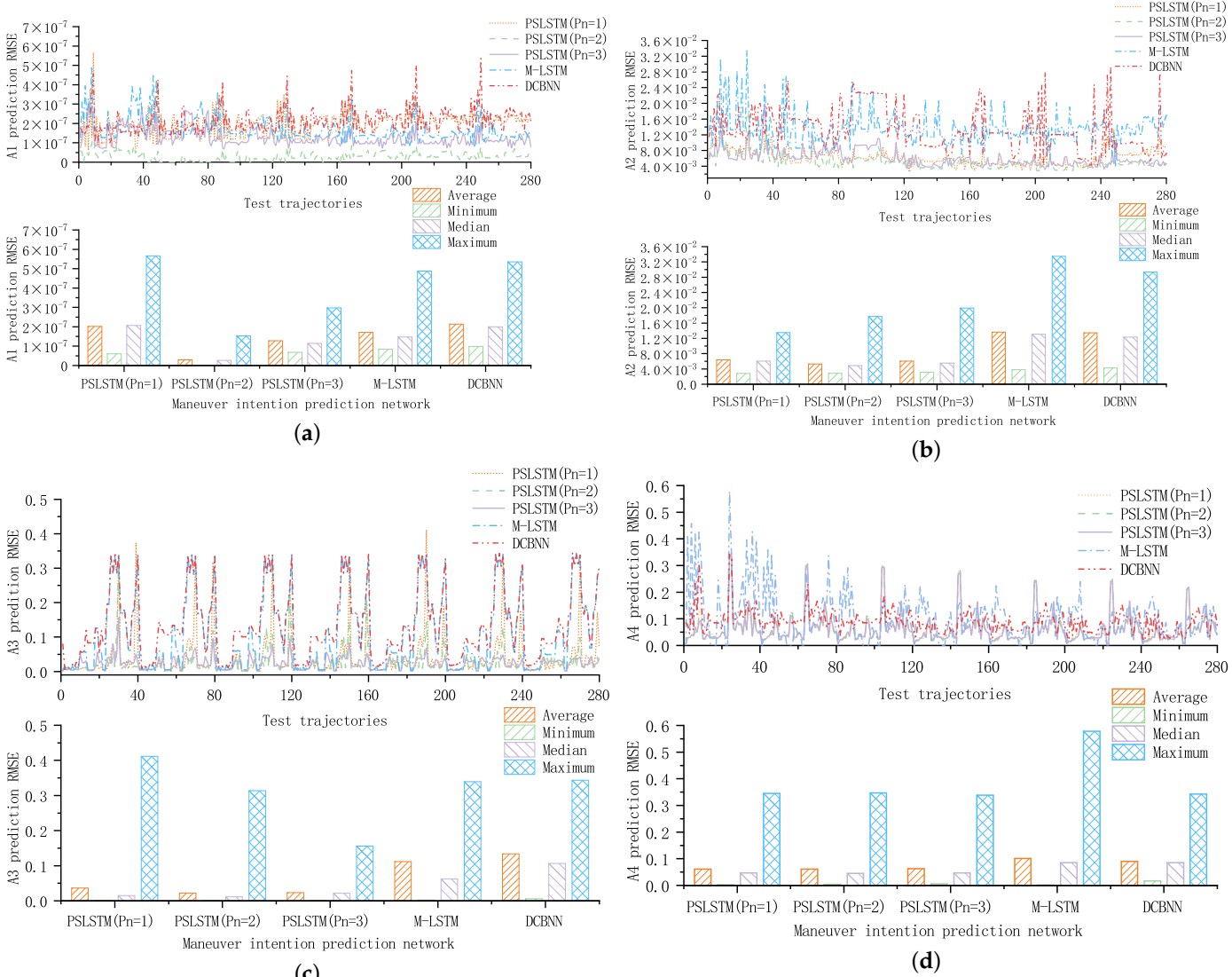

**Figure 12.** Prediction error of the maneuver intention prediction network (*QEG*): (**a**) is the error of A1 prediction, (**b**) is the error of A2 prediction, (**c**) is the error of A3 prediction, and (**d**) is the error of A4 prediction.

In summary, the test results show that compared with the series stacked network, the parallel-stacked neural network has improved the prediction accuracy, especially in the maneuver intention-related prediction of the HGVs. The PSLSTM with $P_n = 3$ can input three consecutive state vectors at the same time, so the network is more focused on

the correlation between short-term data. The PSLSTM with $P_n = 3$ shows advantages in the intention recognition and prediction of $SG$ trajectories. In the PSLSTM with $P_n = 2$, considering two consecutive state vectors, the long-term association in memory also plays a positive role in the results. The PSLSTM with $P_n = 2$ shows more advantages in $QEG$ trajectories intention recognition and prediction.

*4.2. Comparison between SS-LCBBA and Baseline Algorithms*

The SS-LCBBA was used to track three planned typical trajectories. The improved contract network protocol (ICNP) and blackboard model (BM) were chosen as the baseline algorithms. The ICNP is an autonomous coordination mechanism under the dynamic-distributed structure. According to the characteristics of the satellite constellation tracking system, the ICNP inherits the primary process of the traditional contract network protocol (TCNP) and improves it with an alternate tenderer mechanism, self-check procedure, and regret mechanism. The BM is a global consensus algorithm for distributed structures, and the blackboard is a global database used by knowledge sources to communicate results as blackboard entries. For specific details of the ICNP and BM algorithms, please refer to [35]. The observation distance $D_a$ between the satellite, the vehicle in the mission planning process, the impact of the communication delay $E_l$ between satellites, and the computational delay $E_c$ of satellites on the mission planning calculation are considered in the value function.

$D_l$ is calculated as follows:

$$E_l = \frac{\sum\limits_{i=0}^{i} D_t + D_n \cdot n}{C} \tag{29}$$

where $D_n$ is the distance of neighboring satellites in the same orbit, and $n$ is the number of single hops in the same orbit of the link. $D_t$ represents the distances of single hops in the different orbits of the link. $C$ is the speed of light. $E_c$ is calculated as follows:

$$E_c = \sum\limits_{i_s=0}^{i_s} \left( T_c^e - T_c^s \right) \tag{30}$$

where $i_s$ is the satellite serial number, $T_c^s$ is the start time of the planning calculation, and $T_c^e$ is the end time of the planning calculation. The tracking distance evaluation value is as follows:

$$C_L = \frac{u_{if} * c_H}{\sqrt{2\pi}\sigma} \cdot \exp\left( -\frac{\left( D_a - \frac{\sin\left( \pi - \theta - ar\sin\left( \frac{(H_s + R_e) \cdot \sin\theta}{H_n + R_e} \right) \right) \cdot (R_e + H_n)}{\sin\theta} \right)^2}{2\sigma^2} \right) \tag{31}$$

where $u_{if}$ shows whether the target is tracked. $c_H$ is the target value. $H_s$ is the satellite orbit altitude. $\theta$ is the pitch angle of the sensor. $H_n$ is the height of the near space. $\sigma$ is the standard deviation. The value of the inter-satellite link is evaluated as follows:

$$C_i = \frac{u_{if} * c_H}{\sqrt{2\pi}\sigma} \cdot \exp\left( \frac{E_l}{2\sigma^2} \right) \tag{32}$$

The value of planning calculation time is evaluated as follows:

$$C_o = \frac{u_{if} * c_H}{\sqrt{2\pi}\sigma} \cdot \exp\left( \frac{E_c}{2\sigma^2} \right) \tag{33}$$

$\sigma$ is set to $\sigma = \sqrt{\frac{2}{\pi}}$.

The value function is shown below:

$$S = L_l \cdot L_t \cdot (\omega_l \cdot C_L + \omega_i \cdot C_i + \omega_o \cdot C_o) \tag{34}$$

where $L_l$ is the pitch loss factor and $L_t$ is the distance loss factor, which are defined in Section 3.1. $< \omega_l, \omega_i, \omega_o >$ are the weighting factors. As the tracking distance is more important, the weighting factor is set to $< 0.4, 0.3, 0.3 >$.

### 4.2.1. Selection of Typical Trajectories

In this paper, the initial conditions of trajectories are reflected in Table 1. Three typical trajectories of HGVs are reflected in Table 2.

**Table 1.** Initial conditions.

| $\theta$ (deg) | $\phi$ (deg) | $h$ (km) | $v$ (m/s) | $\gamma$ (deg) | $p$ (deg) |
| --- | --- | --- | --- | --- | --- |
| 0 | 0 | 59 | 3060 | $-1$ | 15 |
| 10 | 30 | 56 | 3060 | $-5$ | $-20$ |
| $-10$ | $-30$ | 46 | 2040 | $-4$ | 30 |

**Table 2.** Maneuver type settings.

| | |
| --- | --- |
| Trajectory 1 | HGV does *SG* glide in the longitudinal direction. The normalized lift coefficient for *SG* maneuver is $c_l = 0.7$. It does no lateral evasive maneuver. |
| Trajectory 2 | HGV does *QEG* glide in the longitudinal direction. The normalized lift coefficient for *QEG* maneuver is $c_l = 1.6$. It does lateral evasive maneuvers with the bank angle of $\pm 10°$ degrees at 200~400 s, no maneuvers laterally with the bank angle of $\pm 20°$ degrees at 401~750 s, and lateral evasive maneuvers after 750 s. |
| Trajectory 3 | HGV does *QEG* glide in the longitudinal direction. The normalized lift coefficient for *QEG* maneuver is $c_l = 1.6$. It does lateral turning maneuvers with the bank angle of $-70°$ degrees at 200~600 s, and lateral turning maneuvers with the bank angle of $+70°$ degrees at 600~1200 s, and lateral evasive maneuvers after 1200 s. |

### 4.2.2. Satellite Cluster Architecture Settings

In this paper, the satellite constellation configuration of Walker is defined as $i : t/p/f = 90° : 28/4/2$, where $i$ is the inclination, $t$ is the total number of satellites, $p$ is the number of equally spaced planes, and $f$ is the relative spacing between satellites in adjacent planes. The notation means that the satellite constellation consists of 28 satellites with an orbital inclination of 90 degrees, evenly distributed over four orbital planes, with relative spacing being set to two. The orbital altitude of the satellite constellation is set to 1200 km.

Each satellite carries an infrared sensor. The visual field of the sensor is a cone with an angle of $3°$, and the detection distance of the sensor is $r = 6500$ km. The sensor has two states: standby and tracking. The sensor in the standby state is capable of flying back to the target position at a maximum rotation speed up to $60°/s$. In the tracking state, the sensor is constrained by the imaging stability, so the rotation speed is set to $5°/s$. The range of the sensor pitch angle $[\theta_{\min}, \theta_{\max}]$ is $[\arcsin\left(\frac{R_e}{H_s + R_e}\right), \arcsin\left(\frac{H_n + R_e}{H_s + R_e}\right)]$, where $H_s$ is the satellite orbit altitude, and $H_n$ is the height of near space.

The interstellar link uses laser communication. In order to reduce link loss, it is necessary to avoid crossing the near space, so a minimum link height needs to be considered link height $H_c$. The maximum inter-satellite communication distance can be expressed as:

$$L_r^{\max} = 2\sqrt{(H_s + R_e)^2 - (H_n + R_e)^2} \tag{35}$$

### 4.2.3. The Maneuver Intention Prediction Network Parameter Settings

The intention recognition network uses the network structure shown in Section 2.2, where each stacked layer has the same network structure, with 128 neurons in the fully connected layer (FC) and 512, 256, and 128 neurons in the three LSTM layers. The number of parallel networks $P_n$ is selected according to the recognition accuracy test results and the complexity of the intention parameters, $A_1$. The number of parallel networks for the intention recognition network of the *QEG* trajectory was set to $P_n = 2$. The number of parallel networks for the *SG* trajectory recognition network was set to $P_n = 3$. The joint prediction network used the network structure, and each stacked network had the same structure, with the numbers of neurons in two FC of 256 and 128, in order; and the number of neurons in three LSTM layers of 512, 256, and 128, in order. The number of parallel networks $P_n$ was chosen according to the type of longitudinal maneuver. The length of the input sequence was set to 50.

### 4.2.4. Results

In this paper, simulating and analyzing the coherent tracking process of the satellites is focused on maneuver trajectories. The single-overlay mission planning results of the baseline algorithm and SS-LCBBA algorithm were recorded in the form of Gantt charts. Then, the cumulative tracking score of the value function was used to reflect the improvement of the optimal switching windows on the observation score of the coherent tracking mission, for which the value function was used in Equation (34).

Case 1: Trajectory 1

The task planning results of the SS-LCBBA and the baseline algorithm are shown in Figure 13a. The optimal switching window decision algorithm in the SS-LCBBA provides a more suitable switching period time for the satellite. As a result, the SS-LCBBA obtained a higher score. As shown in Figure 13b, the baseline algorithm chose different satellite sequences, but the scores are relatively close. Hence, the SS-LCBBA could choose a more suitable switching time and effectively suppressed the score decay during satellite switching.

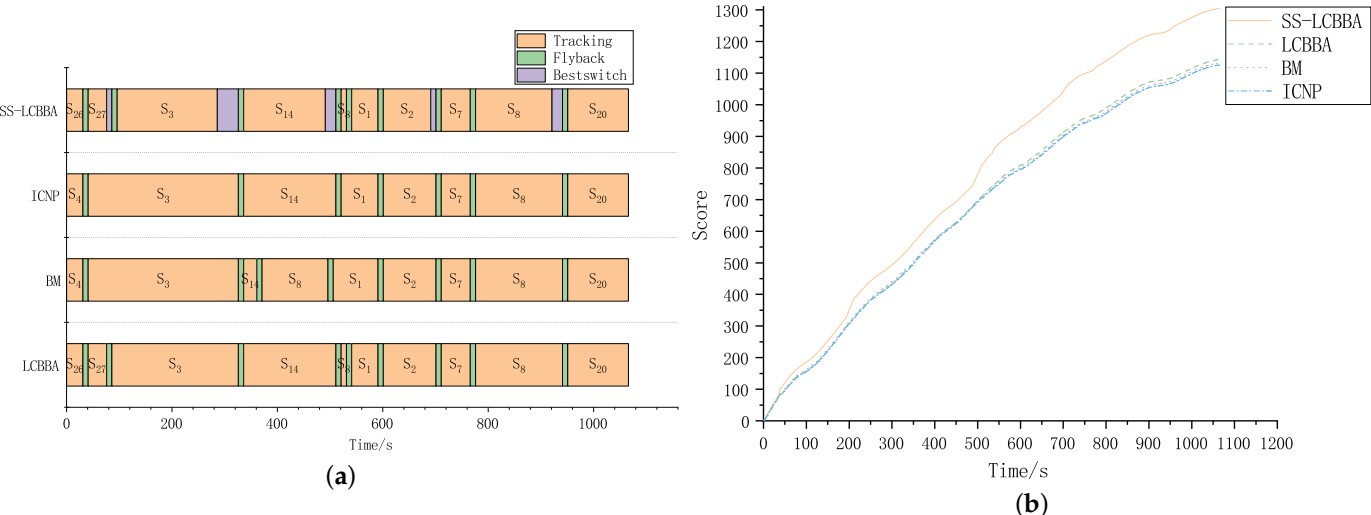

**Figure 13.** The planning results of tracking trajectory 1: (**a**) is a Gantt chart comparing the planning results and (**b**) is the cumulative tracking score of the value function.

Case 2: Trajectory 2

As shown in Figure 14a, the vehicle's longitudinal maneuver was tiny, the flight-path angle of the *QEG* trajectory changed less, and the altitude of the vehicle became a steadily decreasing trend. Even though the vehicle performed evasive maneuvers laterally, the constellation tracking satellites monitored at a higher frequency to achieve full tracking

of the vehicle. The ICNP algorithm has a message center. The planning algorithm prefers satellites with longer tracking times to reduce the computational and communication delays caused by switching satellites. The remaining three mission planning methods are all distributed planning methods. The LCBBA is locally consistent, and its faster response time allows for more satellite switching. Therefore, as shown in Figure 14b, LCBBA and SS-LCBBA selected the satellite sequence with a closer observation distance and obtained higher scores. The SS-LCBBA achieved a higher score when tracking lateral coherent maneuver vehicles.

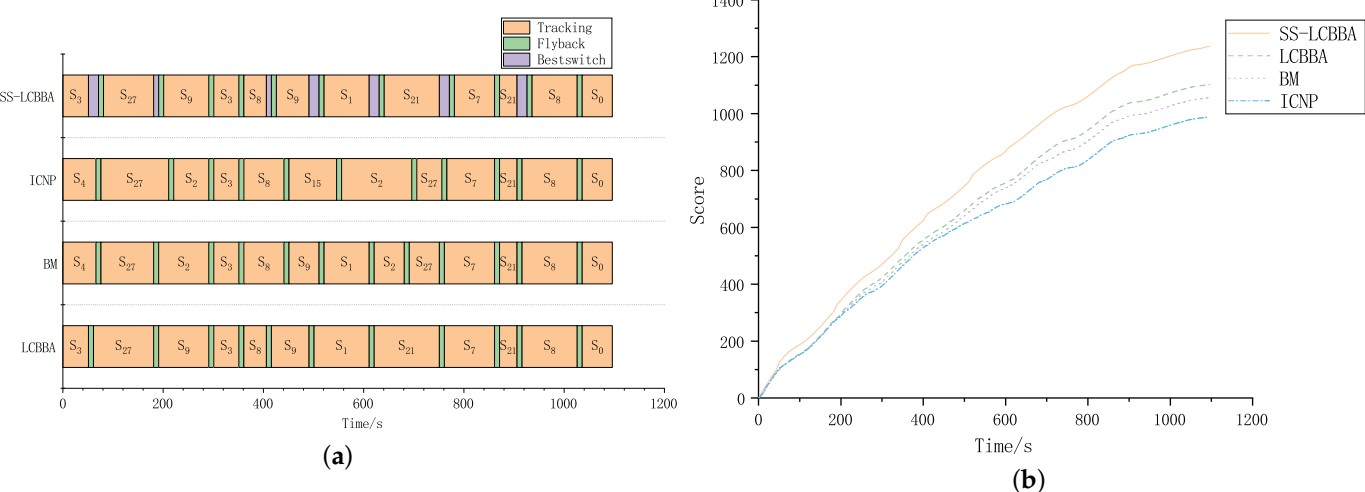

**Figure 14.** The planning results of tracking trajectory 2: (**a**) is a Gantt chart comparing the planning results and (**b**) is the cumulative tracking score of the value function.

Case 3: Trajectory 3

As shown in Figure 15a, the optimal switching window overlaps with the tracking window of the tracking satellite. The reassigned satellite could start tracking the vehicle when the tracking satellite had not reached the tracking limit, given that the observation conditions were better, and the cumulative tracking score was higher. The initial altitude of the vehicle was relatively high, and the constellation had excellent visibility. Therefore, the mission planning algorithm can detect planning results with higher scores faster. As shown in Figure 15b, the four algorithms can obtain relatively close scores in the early stage. Due to the significant lateral maneuver and lower flight altitude in the mid-flight of the vehicle, the mission planning algorithm needs to switch tracking satellites frequently to satisfy the full segment tracking of the vehicle by the constellation.

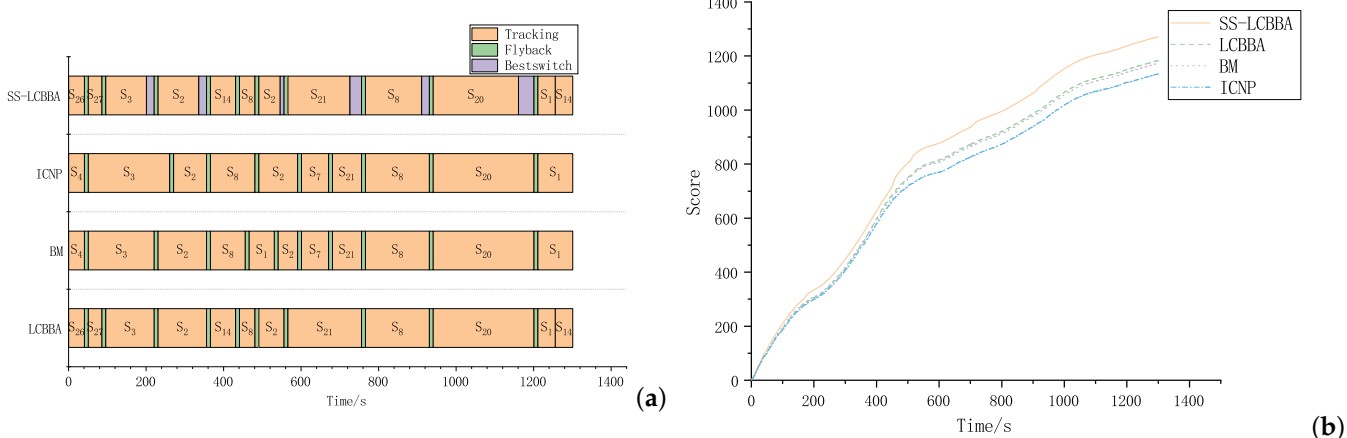

**Figure 15.** The planning results of tracking trajectory 3: (**a**) is a Gantt chart comparing the planning results and (**b**) is the cumulative tracking score of the value function.

## 5. Summary

To address the uncertainty of maneuver intention in tracking HGVs, a quantitative maneuver intention model was established first. Then, a PSLSTM neural network architecture was designed to recognize and predict the maneuver intention based on the stacked LSTM neural network. Finally, the SS-LCBBA was proposed. The loss coefficients were added to the value function of the LCBBA distributed mission planning algorithm, and a switching window selecting algorithm was designed to improve the adaptability of the LCBBA to the uncertainty of the HGV maneuver intention.

The PSLSTM network can still preserve the long-term memory of the data while considering the short-term correlation. The PSLSTM network has better performance for the problem of predicting and recognizing the maneuver intentions of HGVs, as shown in the simulation results. Obtaining the maneuver intentions of HGVs can assist satellites in calculating tracking values and selecting switching times. The value loss factor can prompt the satellite to consider the uncertainty of HGVs' maneuver intentions when selecting targets to track. The sensor-switching strategy can give the satellite a reasonable time period to take over the mission.

The SS-LCBBA algorithm has a higher cumulative tracking score than the baseline algorithms for HGVs. Local consensus algorithms have higher response efficiency for new tasks, but the number of satellite switches increases. Therefore, the sensor-switching strategy is an effective method to improve the tracking coherence during satellite switching. Additionally, considering the uncertainty of the HGV maneuver intention, adding loss coefficients to the value function can effectively enhance the adaptability of the planning algorithm and make the algorithm score higher.

However, the method proposed in this paper also possesses the following limitations: (a) Maneuver intention prediction is limited to the short term and does not allow for long-term prediction. (b) It is assumed that the position data of the HGVs are accurate. However, in practical application scenarios, observation errors usually occur due to the uncertainty of the sensors. (c) The mission planning algorithm was simulated and tested on a civilian computer without considering the onboard hardware-in-the-loop latency.

For future research topics, there are several extension directions worth exploring. (a) Design a method for dynamic selection of prediction results considering the frequency of HGVs' maneuvers. If HGVs have multiple different maneuver intentions within a prediction sequence length, the accuracy of the prediction will be degraded. Therefore, a method needs to be designed to select the effective sequence lengths in the prediction results and dynamically adjust the frequencies of maneuver intention prediction. (b) Design a new multi-satellite cooperative trajectory prediction method. In practical application scenarios, the measurement angle of sensors usually has errors. The state sequences with errors will mislead the deep neural network to output biased results. Therefore, a new multi-satellite collaborative prediction algorithm is needed to improve the accuracy of trajectory prediction through reasonable sensor scheduling and data pre-processing. (c) Design a trajectory correlation algorithm for HGVs, which have high maneuverability and multiple HGVs can change their relative positions frequently during the flight. Therefore, it is necessary to design a trajectory correlation method that can be applied to HGVs to avoid the interference of the tracking algorithm and other algorithms such as the prediction algorithm by observation data of different targets.

**Author Contributions:** Conceptualization, C.W. and Y.C.; methodology, Y.C.; software, Y.W.; validation, Y.C.; investigation, B.Y.; resources, Y.W.; data curation, Y.C.; writing—original draft preparation, Y.C.; writing—review and editing, Y.Z.; visualization, Y.C. All authors have read and agreed to the published version of the manuscript.

**Funding:** This research was funded by the Open Fund of National Defense Key Discipline Laboratory of Micro-Spacecraft Technology (grant number HIT.KLOF.MST.201703).

**Institutional Review Board Statement:** Not applicable.

**Informed Consent Statement:** Not applicable.

**Data Availability Statement:** Not applicable.

**Conflicts of Interest:** The authors declare no conflict of interest.

**Abbreviations**

| HGVs | Hypersonic glide vehicles |
|---|---|
| LSTM | Long and short-term memory |
| PSLSTM | Parallel-stacked long and short-term memory |
| LCBBA | Local consensus-based bundle algorithm |
| SDA | Space development agency |
| SS-LCBBA | The local consensus-based bundle algorithm with the sensor-switching strategy |
| *QEG* | Quasi-equilibrium gliding |
| *SG* | Skip gliding |

## Appendix A. Training Data

*Appendix A.1. Initial Conditions*

In Table A1, $\theta$ and $\phi$ are longitude and latitude, respectively. $h$ is the initial altitude, $v$ is the initial velocity, $\gamma$ is the initial flight-path angle, and $p$ is the initial heading angle.

**Table A1.** Initial conditions.

| $\theta$ (deg) | $\phi$ (deg) | $h$ (km) | $v$ (m/s) | $\gamma$ (deg) | $p$ (deg) |
|---|---|---|---|---|---|
| 0 | 0 | 60 | 3060 | −2 | 0 |
| 30 | 10 | 58 | 3060 | −1 | 60 |
| 60 | 20 | 56 | 3060 | −3 | 120 |
| 120 | 30 | 54 | 3060 | −5 | 180 |
| 150 | 40 | 52 | 3060 | −1 | −60 |
| 180 | 50 | 50 | 3060 | −3 | −120 |
| 210 | 60 | 55 | 2040 | −4 | 0 |
| 240 | −10 | 53 | 2040 | −5 | 60 |
| 270 | −20 | 51 | 2040 | −1 | 120 |
| 300 | −30 | 49 | 2040 | −2 | 180 |
| 330 | −40 | 47 | 2040 | −4 | −60 |
| 360 | −50 | 45 | 2040 | −3 | −120 |

*Appendix A.2. Longitudinal Maneuver Parameters*

In Table A2, $c_l$ is the normalized lift coefficient, described as $c_l = \frac{C_L}{C_L^*}$, where $C_L^*$ is the lift coefficient at the maximum lift-to-resistance ratio. It is described as $C_L^* = \sqrt{\frac{C_{D0}}{K}}$, where $C_{D0}$ and $K$ are the HGV's aerodynamic parameters independent of Mach. We combined two longitudinal maneuver types, *QEG* and *SG*, with seven $c_l$ for a total of 14 longitudinal maneuver types.

**Table A2.** Longitudinal maneuver parameters.

| *QEG/SG* | | | | | | |
|---|---|---|---|---|---|---|
| $c_l$ | 0.4 | 0.7 | 1.0 | 1.3 | 1.6 | 1.8 | 2.0 |

*Appendix A.3. Lateral Maneuver Parameters*

In Table A3, HGVs maintain a fixed bank angle during flight. $\sigma$ is the bank angle of the HGV.

**Table A3.** Lateral maneuver parameters.

| | Velocity Inclination Angle | | | | | | | | | | |
|---|---|---|---|---|---|---|---|---|---|---|---|
| $\sigma$ | $0°$ | $15°$ | $-15°$ | $30°$ | $-30°$ | $45°$ | $-45°$ | $60°$ | $-60°$ | $75°$ | $-75°$ |

In Table A4, HGVs adjust $\sigma$ at a fixed time period. $t$ is the flight time of HGVs.

**Table A4.** Lateral maneuver parameters.

$\sigma = 0°, 0 \leq t < 550; \sigma = 15°, 550 \leq t < 1200; \sigma = 0°, 1200 \leq t;$
$\sigma = 0°, 0 \leq t < 550; \sigma = -15°, 550 \leq t < 200; \sigma = 0°, 1200 \leq t;$
$\sigma = 0°, 0 \leq t < 500; \sigma = 30°, 500 \leq t < 1100; \sigma = 0°, 1100 \leq t;$
$\sigma = 0°, 0 \leq t < 500; \sigma = -30°, 500 \leq t < 1100; \sigma = 0°, 1100 \leq t;$
$\sigma = 0°, 0 \leq t < 450; \sigma = 45°, 450 \leq t < 1000; \sigma = 0°, 1000 \leq t;$
$\sigma = 0°, 0 \leq t < 450; \sigma = -45°, 450 \leq t < 1000; \sigma = 0°, 1000 \leq t;$
$\sigma = 0°, 0 \leq t < 400; \sigma = 60°, 400 \leq t < 900; \sigma = 0°, 900 \leq t;$
$\sigma = 0°, 0 \leq t < 400; \sigma = -60°, 400 \leq t < 900; \sigma = 0°, 900 \leq t;$
$\sigma = 0°, 0 \leq t < 350; \sigma = 75°, 350 \leq t < 800; \sigma = 0°, 800 \leq t;$
$\sigma = 0°, 0 \leq t < 350; \sigma = -75°, 350 \leq t < 800; \sigma = 0°, 800 \leq t;$
$\sigma = 0°, 0 \leq t < 100; \sigma = -15°, 100 \leq t < 550; \sigma = 15°, 550 \leq t < 1100;$
$\sigma = 0°, 0 \leq t < 100; \sigma = 15°, 100 \leq t < 550; \sigma = -15°, 550 \leq t < 1100;$
$\sigma = 0°, 0 \leq t < 150; \sigma = -30°, 150 \leq t < 550; \sigma = 30°, 550 \leq t < 1050;$
$\sigma = 0°, 0 \leq t < 150; \sigma = 30°, 150 \leq t < 550; \sigma = -30°, 550 \leq t < 1050;$
$\sigma = 0°, 0 \leq t < 200; \sigma = -45°, 200 \leq t < 550; \sigma = 45°, 550 \leq t < 1000;$
$\sigma = 0°, 0 \leq t < 200; \sigma = 45°, 200 \leq t < 550; \sigma = -45°, 550 \leq t < 1000;$
$\sigma = 0°, 0 \leq t < 250; \sigma = -60°, 250 \leq t < 550; \sigma = 60°, 550 \leq t < 950;$
$\sigma = 0°, 0 \leq t < 250; \sigma = 60°, 250 \leq t < 550; \sigma = -60°, 550 \leq t < 950;$
$\sigma = 0°, 0 \leq t < 300; \sigma = -75°, 300 \leq t < 550; \sigma = 75°, 550 \leq t < 900;$
$\sigma = 0°, 0 \leq t < 300; \sigma = 75°, 300 \leq t < 550; \sigma = -75°, 550 \leq t < 900;$

In Table A5, HGVs continuously switch $\sigma$ at fixed intervals. $T$ is the switching interval of $\sigma$.

**Table A5.** Lateral maneuver parameters.

| $\sigma$ | $\pm10°$ | $\pm10°$ | $\pm15°$ | $\pm15°$ | $\pm30°$ | $\pm30°$ | $\pm30°$ | $\pm45°$ | $\pm45°$ |
|---|---|---|---|---|---|---|---|---|---|
| $T$ | 100 s | 150 s | 100 s | 150 s | 100 s | 80 s | 50 s | 50 s | 30 s |

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
