# Peer review of "Intention Prediction of a Hypersonic Glide Vehicle Using a Satellite Constellation Based on Deep Learning"

_mathematics, doi:10.3390/math10203754_

Round 1

Reviewer 1 Report

This is a well-presented work, however, I would like to point out a couple of things:

1) I would like the authors to add training loss plots to show that the algorithm/neural network is indeed properly trained and converges.

2) Provide more motivation in the introduction and present a more high-level explanation as to why the chosen architecture PSLSTM network and SS-LCBBA algorithm perform better.

 3) There are typos here-and-there and missing spaces.

Reviewer 2 Report

The paper presents an intention prediction algorithm for hypersonic glide vehicles using machine learning-based methods. Particularly, it prosed PSLSTM neural net architecture to recognize and predict the maneuver intention.

The abstract could also mention the results and what was observed concisely.

Line 5-6: “A HGVs 5maneuver intention model, which not only describes the types of motions of HGVs in the longitudinal and lateral directions but also quantifies the magnitude and frequencies of motions is designed.”

This does not sound correct. Please check.  Ex: “motions are designed”, “An HGVs maneuver”

The paper can be further improved by clarifying a little bit more of the introduction. The introduction covers a lot of literature but could be better by explaining Section 1.2 more. For example, the “probability transfer matrix” appears very abruptly. The authors need to write from the point of the readers.

Technically, the paper is mathematically sound. However, I would like to know when the proposed algorithm failed. Is it always able to predict the maneuvers? What happens if there is uncertainty in the sensors?

What are some of the downsides of the proposed work? And how can you improve it in the future? 

Reviewer 3 Report

1. Abstract should be rewritten focussing on the problem statement and proposed work.

2. Introduction section needs to be re-written to improve its quality and readability.

3. Clearly explain on the novelty and complexity of this research or method

4. Provide a comprehensive comparison of performance with other recent works conducted and published

5. Why LSTM is used in this study? Justify

6.  In Section 2.2 Pre-trained models and transfer learning. Explain in detail the pre-trained models.

7.  Include future work in the conclusion section.

8.  Check the writing with a native English speaker for improving readability

Reviewer 4 Report

This work applied deep learning method for intention prediction of satellite constellation HGV problems. The manuscript is written with sufficient background, solid design, algorithm description, results presentation. More details of the experiment and simulations can be further provided. Before proceeding to publication, please address a few minor comments below. Also, the English must be improved. Grammar mistakes can be found throughout the manuscript. Please revise and/or check with English editor(s).

1.      [Section 1.4. Research content] The authors summarized the main work in this section, however, the actual advantage compared to prior arts and major contribution to the field should be further clarified too. For examples, any quantified performance contributions from the new dynamic model or neural network prediction?

2.    [Equation 2]. Please explicitly define rate of change ()

3.      [Line 227] Please properly fix the format issue in the equation embedded in lines.

4.      [Section 4.1. Simulation and analysis of networks] What is the software platform and programming language used for the simulation development? Was any simulation conducted in real-time using the trained ML-based prediction model with consideration of hardware-in-the-loop latency?

Round 2

Reviewer 3 Report

The research topic is very interesting and meaningful. There are some comments as follows.

1. In the Introduction, the contribution needs to be streamlined to stress the main points. 

2.  A good abstract should have the following elements (a) Introduction/ Background (b) Method (c) Results (d) Discussion/ Conclusion.?

3. Please avoid long sentences.

4.  The root means square error (RMSE) is used in this paper. The origin references should be provided. 

5.  The format of the references should be unified.

6. Future research topics are suggested to be provided in the conclusion part.
